# Comparative Analysis of Blood Transfusion Accuracy and Hemolysis Rate of Transfusion Cartridge Set Between Conventional Infusion Pumps and Cylinder-Type Infusion Pumps

**DOI:** 10.3390/biomedicines12112421

**Published:** 2024-10-22

**Authors:** Hee-Young Lee, Sun-Ju Kim, Kang-Hyun Lee, Il-Hwan Park, Hyeok-Jin Jeon, Hyun Youk

**Affiliations:** 1Department of Emergency Medicine, Yonsei University Wonju College of Medicine, Wonju 26426, Gangwon State, Republic of Korea; hylee3971@yonsei.ac.kr (H.-Y.L.); crescendo@yonsei.ac.kr (S.-J.K.); ed119@yonsei.ac.kr (K.-H.L.); 2Regional Trauma Center, Wonju Severance Christian Hospital, Wonju 26426, Gangwon State, Republic of Korea; nicecs@yonsei.ac.kr; 3Department of Emergency Medical Technician, Gyeongbuk Provincial College, Yechoen 36830, Gyeongsangbuk-do, Republic of Korea; ohooms@hanmail.net

**Keywords:** infusion pump, cylinder type, hemolysis, blood transfusion

## Abstract

Background: Infusion pumps are critical in delivering fluids, including medications and blood products, in controlled amounts. However, conventional pumps can cause hemolysis and other issues such as flow variations and infection risks, especially during blood transfusions. To address these limitations, a novel cylinder-type infusion pump, the Anyfusion H-100, was developed, which includes a specialized blood transfusion cartridge set that combines syringe and peristaltic infusion methods. This study evaluates the accuracy and hemolysis rates of the Anyfusion H-100 compared to conventional pumps, aiming to confirm its viability as a safe and effective medical device for blood transfusions. Methods: This study evaluated six different infusion rates (10–180 cc/hr) and conducted 57 transfusion trials, 20 of which used a 1:1 blood–saline dilution. Blood transfusion accuracy was measured using research-grade packed red blood cells, and hemolysis rates were assessed before and after transfusion by chi-square tests and independent sample *t*-tests. Results: Anyfusion demonstrated an average transfusion error rate of 3.77%, compared to 4.00% for the Terufusion, with no statistically significant difference in hemolysis rates (*p* = 0.697). Bland–Altman plots confirmed their equivalent performance, with hemolysis rates of 0.566 ± 0.095% for Anyfusion and 0.518 ± 0.126% for Terufusion. Conclusions: Anyfusion provides an accurate and reliable blood transfusion performance comparable to that of Terufusion, with no significant difference in hemolysis rates; its integration of syringe and infusion methods shows a potential for safer and more efficient transfusion practices, especially in pediatric and emergency settings.

## 1. Introduction

Infusion pumps are medical devices that are used to deliver fluids, such as nutrients and medications, into a patient’s body in controlled amounts [1]. The evolution of infusion pumps from basic mechanical devices in the 1970s and 1980s to the advanced, multifunctional devices of today is well documented [2]. Modern infusion pumps incorporate various features and alarms that are designed to alert users about issues such as nearing the completion of infusions or the need for attention due to detected anomalies [3].

Infusion pumps for medications are essential devices that are used in intensive care units, emergency rooms, and neonatal units. They are employed when there is a need to administer precise or large amounts of medication to patients or to deliver an accurate volume of drugs per hour. In particular, when administering red blood cells and platelets to pediatric patients (newborns), only a small volume of blood is required, and due to their small body size, there is a risk of circulatory volume overload. Therefore, it is crucial that the correct amount is transfused at a consistent rate [4,5]. In addition to pediatric patients, blood components such as red blood cells, plasma, and platelets are applied according to the therapeutic needs of the patient. Additionally, in cases of massive hemorrhage due to trauma, whole blood may be used [6].

Generally, infusion pumps for medications inject a predetermined volume into the patient based on a program that rotates a motor. When using an infusion pump for blood transfusions, a constant pressure is applied, which may cause physical damage to red blood cells as they pass through the needle, increasing the risk of hemolysis [7]. Therefore, when transfusing red blood cells or other blood products using an infusion pump, it is crucial to verify whether the device is suitable for its intended purpose, especially since visible hemolysis can occur with red blood cell products [8]. In the case of syringe-based infusion pumps, frequent syringe replacements pose a risk of infection, necessitating careful attention and leading to increased workload. Additionally, issues associated with the operation of traditional infusion pumps, such as infusion accuracy, backflow, and startup delays, can become problematic during low-volume transfusions. Even with this issue, small dosing errors in these vasoactive drugs can result in severe hemodynamic fluctuations in vulnerable patients [9,10]. Conventional infusion pumps control flow through the peristaltic motion of tubing, which can result in flow variations of ±5% to 20% per hour, depending on the tubing material and elasticity [11]. When issues like tubing recoil arise, the system may need to be reset or the tubing may need to be replaced. Furthermore, the drive mechanism that compresses the tubing may also lead to hemolysis. The syringe pumps used for precise infusion have compatibility issues due to varying syringe specifications among manufacturers, and they require frequent resetting when large volumes are injected. The process of transferring blood into a syringe for precise infusion increases vulnerability to contamination and infection [4,12].

To improve the limitations of the structure of conventional peristaltic and syringe-type infusion pumps, a cylinder-type infusion pump (Anyfusion H-100) was developed that integrates the syringe infusion method with the infusion method (Figure 1). This device operates on the principle of two pistons being attached to two disks within a donut-shaped cylinder that rotates through two motors to aspirate and dispense liquids (drugs), gasses, and fluid solids (food and blood), regardless of gravity or position [13].

In this study, using research-grade blood products, we tested if the blood could be infused at an accurate volume and rate, and examined hemolysis levels before and after blood transfusion to confirm its viability for transfusion. Through this, we aimed to demonstrate that this new type of infusion pump has no functional differences compared to conventional infusion pumps and to prove its suitability as a medical device.

## 2. Materials and Methods

The experiment was conducted with the following six different infusion rates: 10 cc per hour for 3 h, 15 cc per hour for 2 h, 30 cc per hour for 1 h, 60 cc per hour for 1 h, 120 cc per hour for 1 h, and 180 cc per hour for 1 h. Each experiment was repeated six times. Additionally, 20 tests were performed using a 1:1 dilution method with normal saline to simulate emergency blood transfusions. Normal saline was chosen because it prevents hemolysis, provides volume expansion, and is fully compatible with blood products, making it a suitable choice for rapid transfusion under emergency conditions [14]. All experiments were carried out with an experimental group (Anyfusion H-100) and a control group (Terufusion TE-LM700).

### 2.1. Mechanism of Cylinder-Type Infusion Pumps

The cylinder-type infusion pump has two unique characteristics in terms of flow accuracy and stability, which arise from using a dedicated cylinder cartridge including a new operating principle using high-precision motor control and an automatic locking system to completely fix the cartridge to the pump body. The core mechanism of the device is the continuous cross-cycling of two pistons inside the dedicated donut-shaped cylinder cartridge, with independent motor control for each piston. This mechanism converts the linear motion of one piston inside a conventional syringe into the rotary motion of two pistons inside a spherical cylinder. In the donut-shaped cylinder, each piston is programmed to rotate in a different direction from the others, allowing for the precise control of the fluid infusion. The piston between the fluid inlet and outlet has a fixed shaft (S or S′), while the other piston has a rotating shaft (R or R′). Under the precise control of an independent motor according to a set injection rate, the rotating pistons (R or R′) rotate counterclockwise, so that a precise amount of fluid is simultaneously aspirated from the inlet and expelled from the outlet (Figure 2).

### 2.2. Eqiupment and the Packed RBC

The Anyfusion H-100 (Meinntech Corporation, Anyang, Republic of Korea), which combined a syringe and an infusion pump into a single pump unit, was used as the investigational device. Also, Terufusion TE-LM700 (Terumo Medical Corporation, Tokyo, Japan), which uses a syringe-type pump equipped with a transfusion set with a diameter of 4.0 mm, was used as the comparator (Figure 3). This study was also carried out with approval from the Research Ethics Committee of Yonsei University Wonju Severance Christian Hospital (IRB approval number: CR320345).

Blood safety center-certified research-grade blood from the Korean Red Cross (blood that was collected for transfusion but discarded due to slight increases in factors such as AST) was supplied to the transfusion cartridge. For the clinical trial, the blood was stored in a temperature-controlled storage device designed for blood. Blood tests, including CBC and electrolyte tests, were conducted. SST tubes were stored at room temperature for 30 min before being centrifuged at 2500–3000 RPM for 10 min and then refrigerated. EDTA tubes were mixed immediately and refrigerated, with tests requested within one day of sample collection. In cases of sample ineligibility, additional test results were verified through point-of-care testing on the stored samples (Figure 4).

### 2.3. Study Protocols

The values measured between the two devices were compared using the research blood products provided as follows:i.Quantification and error rate according to infusion rate and time;ii.Degree of hemolysis according to infusion rate.

In the non-dilution trials, the blood sample infusion volume per hour was set to 10 cc/hr (180 min), 15 cc/hr (120 min), 30 cc/hr (60 min), 60 cc/hr (60 min), 120 cc/hr (60 min), and 180 cc/hr (60 min). In addition, in the dilution trials, the blood sample infusion volume per hour was set to 60 cc/hr (120 min), 120 cc/hr (60 min), 180 cc/hr (40 min), and 240 cc/hr (30 min) (Figure 5).

### 2.4. Statistical Analysis

In this study, the choice between parametric and non-parametric statistical methods is determined by conducting normality tests on each variable based on the test results. For additional data analysis, the Windows-based statistical software SPSS Ver. 25.0 (SPSS Inc., Chicago, IL, USA) was used. The test results (such as volume, rate, and comparison with the control group) are presented as descriptive statistics, followed by further analysis. In the event of missing data, the list-wise deletion method is applied.

Errors occurring during blood transfusion were verified for differences between the two medical devices using chi-square tests, and the hemolysis rate was verified for statistical significance using independent sample *t*-tests. In addition, the performance equality of the two medical devices was verified using Bland–Altman plots.

## 3. Results

### 3.1. Overall Comparison by Infusion Rate of Each Device

A total of six trials were conducted for each experiment, along with 20 trials using saline-mixed dilutions. During the experiments, one device error occurred with the Anyfusion H-100, leading to one additional trial, resulting in a total of 57 trials. The comprehensive comparison table of average error rates for each infusion rate is shown in (Clinical Trial Comprehensive Comparison Table 3). The average volume was not included, as the volume varied by infusion rate. The average error rate was 3.86% for Anyfusion H-100 and 3.98% for Terumo TE-LM700, with both groups showing an error rate of less than 4%. After the clinical trials, the average blood temperature was 25.1 °C for the Anyfusion H-100 and 24.6 °C for the Terumo TE-LM700 control group, showing a difference of 0.5 °C (Table 1).

### 3.2. Comparison of Error According to Infusion Rate

The comparison of issues according to infusion rate, excluding the dilution test, is shown in Table 2. In low-speed transfusions at 10 cc/hr and 15 cc/hr, visually observable clot formation occurred in both groups with four cases each, accounting for 10.8%. However, at infusion rates of 30 cc/hr or higher, no visually observable clot formation was detected in either group. In the Anyfusion H-100, air was detected in eight cases (21.6%), sample ineligibility in four cases (10.8%), and one error (2.7%). In the Terumo control group, air was detected in six cases (16.2%), sample ineligibility in 8.1%, and no errors occurred. Sample ineligibility occurred due to micro-clot formation or when the research blood had been collected for an excessively long period. It should be noted that the data may vary when transfusing blood to actual patients as opposed to research samples.

### 3.3. Transfusion Volume Error Rate and Laboratory Temperature for Each Experiment

A total of 57 clinical trials were conducted (37 non-dilution trials and 20 dilution trials). The final transfusion error rate for the Anyfusion H-100 was 3.77%, while the Terumo TE-LM700 had a final error rate of 4.00%, indicating a slightly higher accuracy for the Anyfusion H-100 compared to the Terumo TE-LM700. When comparing the error rates of the 37 non-dilution trials to the total trials, the following results were observed:i.Anyfusion H-100: non-dilution: 3.86%; dilution: 3.60%; overall: 3.77%;ii.Terufusion TE-LM700: non-dilution: 3.98%; dilution: 4.05%; overall: 4.00%.

After the final transfusions, the temperatures were 25.0 °C for Anyfusion H-100 and 24.6 °C for Terumo TE-LM700, showing a temperature difference of approximately 0.4 °C (Table 3).

### 3.4. Overall Experiment Error Rate

In Table 4, out of a total of 56 trials, air trapping was reported in 9 cases (15.8%) for Anyfusion H-100 and in 7 cases (12.3%) for Terumo TE-LM700. Clots occurred in four cases (7.0%) each. In the chi-square test, the X^2^ value was 0.294 and the *p* value was 0.863, failing to confirm statistical significance.

### 3.5. Comparison of the Mean Hemolysis Rates of Each Medical Device

Excluding the 20 diluted cases, among the 37 cases analyzed using the *t*-test, the hemolysis rates by speed were 0.566 ± 0.095 for the Anyfusion H-100 and 0.518 ± 0.126 for the Terumo TE-LM700. The t-value was −0.745, and the *p*-value was 0.697, indicating no statistical significance, which suggests that there is no functional difference between the two infusion pumps (Table 5).

In addition, it also used the Bland-Altman plot to determine the distribution of hemolysis rates for the two medical devices (Figure 6). In the figure, the single thick line indicates the mean value, and the dashed line indicates the mean difference ± standard deviation. Most of the difference values between the two devices were distributed within the dashed line range, which means that the two medical devices showed the same performance during transfusion.

## 4. Discussion

The electromechanical infusion device can deliver liquid preparations at a constant rate and can accurately measure the amount infused, making it a common tool that is used for pediatric patients or those with chronic kidney disease. In clinical settings, various commercially available infusion devices are utilized, and factors such as osmotic pressure of drugs or blood products, infusion volume, medication administration procedures, consistency, accuracy, and infusion rate should all be considered to enhance effective patient management and reduce wasted medication [15,16,17].

The error rate for the infusion rate of a typical intravenous pump is reported to be less than ±5%. In addition, the error rate for the infusion rate of an elastic infusion pump is reported to be less than ±15% [18]. The Meintech device, using a cylinder-type medication infusion method with a blood transfusion cartridge, demonstrated a slightly better error rate in time–speed–volume comparisons at 3.77% compared to the control group—Terufusion TE-LM700—which had an error rate of 4.00%. When compared to the error rates presented by manufacturers for drug or fluid infusion (Terumo: ±5%; Baxter: ±7%), Terumo maintained an error rate within 5% for all samples except for two blood specimens, while Baxter had no blood specimens within the error rate. Compared to the error rate standards for infusion pumps, Terumo’s average error rate for infusion speed was below 5%, indicating that it could reliably deliver the set volume to patients. Although this evaluation focused on medication, there was no significant difference in accuracy when compared to the existing literature [18]. However, due to the difference in viscosity between blood and intravenous fluids, it is challenging to apply the same standards to both. Additionally, this study used blood that was diluted with fresh frozen plasma, but the viscosity of concentrated red blood cells used for transfusions is higher, which may lead to a greater actual error.

The hemolysis rate showed little difference, at 0.52% compared to 0.57%, suggesting that this device could serve as a viable alternative to the widely used Terufusion TE-LM700. However, the incidence of air trapping was slightly higher, at 15.8% compared to 12.2% for the Terufusion TE-LM700. If this issue is addressed, the device could be utilized as a more advanced medical device. The degree of damage to red blood cells during infusion is influenced by various factors, including the storage duration of blood products after collection, the diameter of the needle used, and the pressure applied to maintain a constant infusion rate. However, the most significant factors are the characteristics and operating methods of the infusion device itself [19,20,21].

It has been reported that peristaltic pumps are associated with hemolysis. Among 13 existing studies, hemolysis was observed in 10 studies using peristaltic infusion devices. In contrast, volumetric infusion devices that use a cassette mechanism showed no association with hemolysis in 4 out of 6 studies [22]. In a study conducted in South Korea comparing the degree of hemolysis during pediatric transfusions based on infusion devices, no increase in plasma hemoglobin was observed when whole blood was administered. However, an increase in plasma hemoglobin was noted when concentrated red blood cells were infused, although this change was not statistically significant [23].

Based on the results of this study and a review of previous research, it is determined that the Anyfusion H-100 can sufficiently fulfill its role as an infusion pump capable of performing blood transfusions. In further research, it may be suggested to conduct additional clinical trials on the topic of clinical benefits that utilize the advantages of this cylinder pump to reduce hemodynamic instability in patients or reduce the clinical workload of medical staff.

## 5. Conclusions

This study aims to verify whether a device that integrates the syringe infusion method with the infusion method can deliver accurate volumes and rates with the same performance as other infusion pump devices, and whether there is any statistically significant differences in hemolysis rates, thereby demonstrating the potential applicability of this new type of infusion pump. When comparing the Anyfusion H-100 and the Terufusion TE-LM700 using research blood samples, no statistically significant differences were observed in transfusion volume, errors, or hemolysis amounts, indicating equivalent performance. The Bland–Altman plot also showed that the distribution of hemolysis rates mostly fell within the acceptable range, confirming no functional differences between the devices.

However, this study has limitations in that it was a laboratory study using research-grade blood products and safety was not assessed as no actual patients were transfused using these devices. Nevertheless, the Anyfusion H-100, as an infusion pump capable of performing blood transfusions, is expected to be highly useful as a medical device that integrates the syringe infusion method and the infusion method, effectively addressing each method’s functional issues.

## Figures and Tables

**Figure 1 biomedicines-12-02421-f001:**
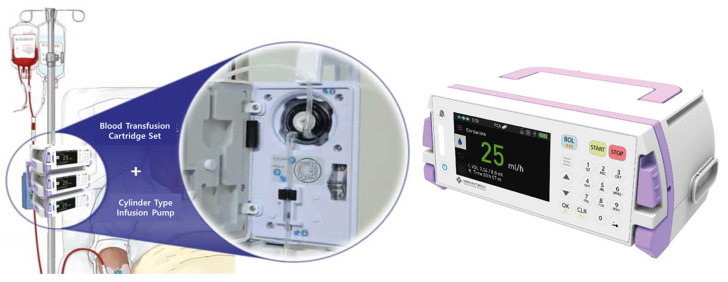
Photo of a cylinder-type infusion pump and transfusion cartridge set.

**Figure 2 biomedicines-12-02421-f002:**
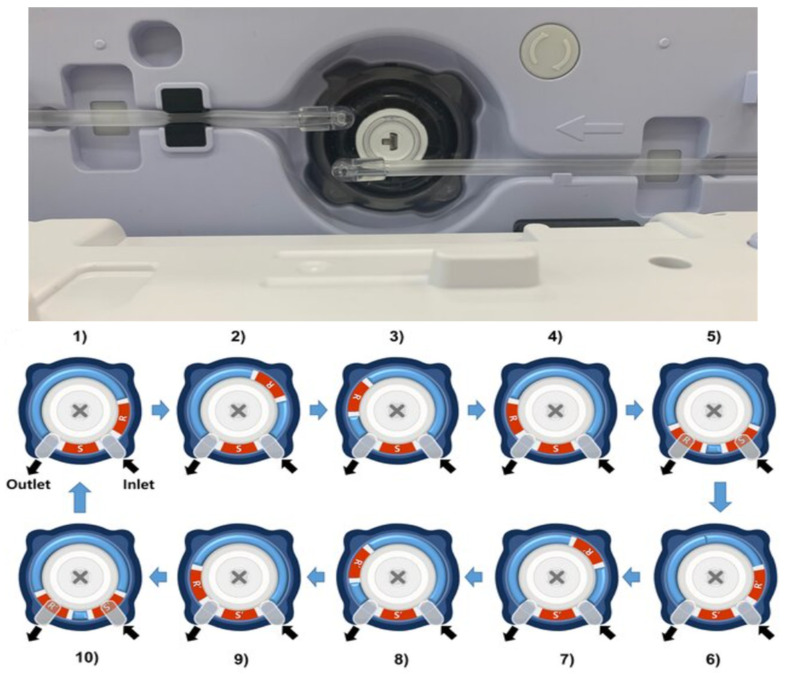
Photo of a cylinder cartridge fixed to the infusion pump body with an automatic locking system and mechanism of cylinder-type pump.

**Figure 3 biomedicines-12-02421-f003:**
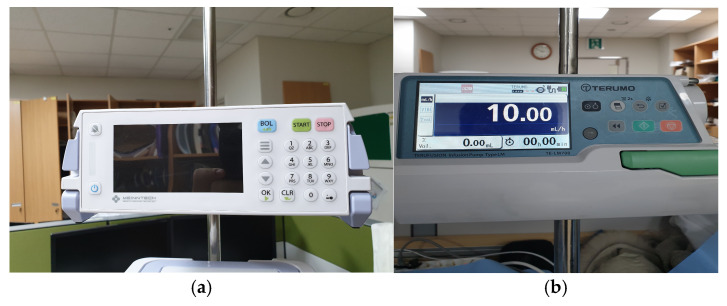
Infusion pump medical devices. (**a**) Test device (Anyfusion H-100); (**b**) comparator (Terufusion TE-LM700).

**Figure 4 biomedicines-12-02421-f004:**
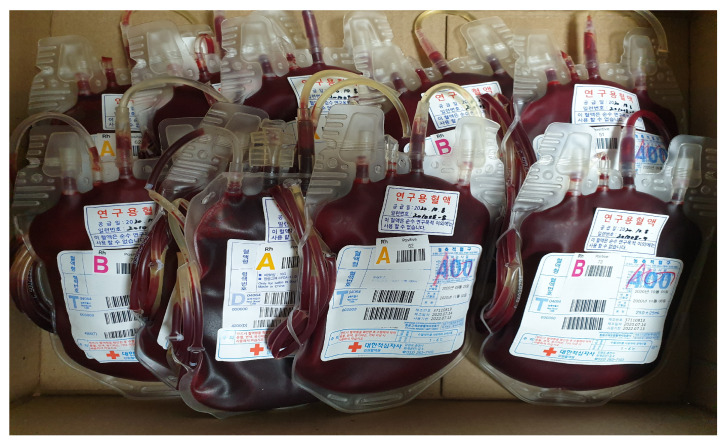
Research-grade blood products supplied by Korean Red Cross Blood Services.

**Figure 5 biomedicines-12-02421-f005:**
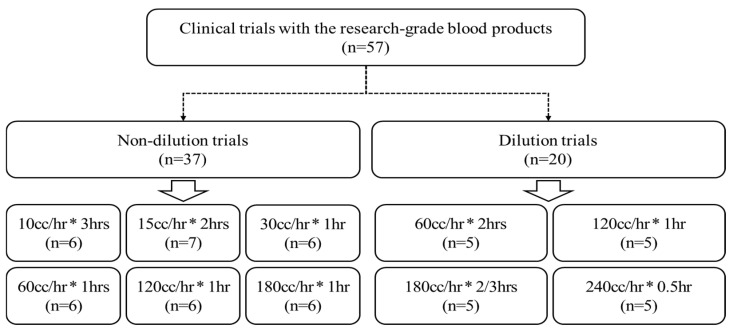
Flowchart of this study using the research-grade blood products for comparative analysis of blood transfusion accuracy and hemolysis rate.

**Figure 6 biomedicines-12-02421-f006:**
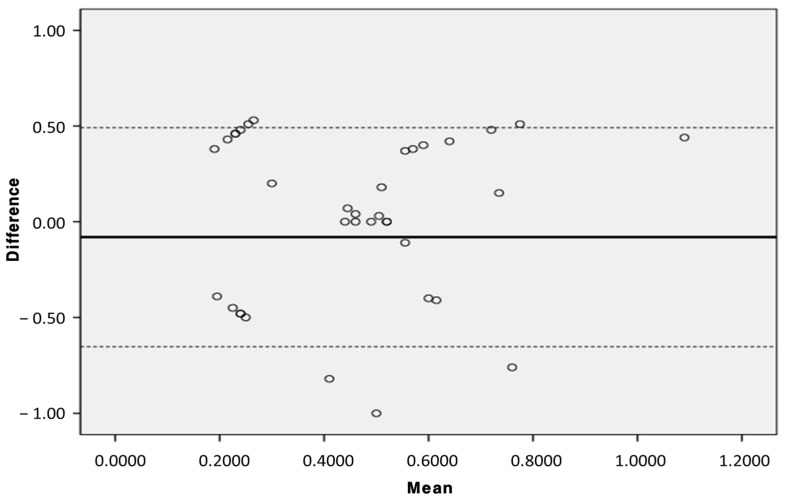
Agreement of hemolysis rates between two medical devices using Bland–Altman plot.

**Table 1 biomedicines-12-02421-t001:** Total infusion volume and error rate by infusion rate.

No.	Infusion Speed (cc/hr)	Duration (hrs)	Anyfusion H-100			Terumo TE-LM700		
			Total Infusion Volume (cc)	Error Rate (%)	Temperature (°C)	Total Infusion Volume (cc)	Error Rate (%)	Temperature (°C)
1	10	3	30.9	4.39%	25.1	28.8	5.00%	24.5
2	15	2	29.5	3.71%	24.9	29.6	3.99%	24.4
3	30	1	31.2	4.28%	24.8	29	3.33%	24.5
4	60	1	58.9	3.92%	24.8	57.2	4.65%	24.6
5	120	1	118.9	4.12%	25.4	120.2	3.33%	24.7
6	180	1	181.6	2.78%	25.4	179.4	3.55%	24.7

**Table 2 biomedicines-12-02421-t002:** Error types and occurrence rates.

No	Infusion Speed (cc/hr)	Duration (hrs)	Anyfusion H-100		Terumo TE-LM700	
			Types of Errors	Number of Occurrences (n, %)	Types of Errors	Number of Occurrences (n, %)
1	10	3	Air	2 (33.3)	Air	1 (16.7)
			Clot	2 (33.3)	Clot	2 (33.3)
			Sample ineligibility	0 (0)	Sample ineligibility	0 (0)
2	15	2	Air	1 (14.3)	Air	1 (14.3)
			Clot	2 (28.6)	Clot	2 (28.6)
			Sample ineligibility	0 (0)	Sample ineligibility	0 (0)
3	30	1	Air	1 (16.7)	Air	1 (16.7)
			Clot	0 (0)	Clot	0 (0)
			Sample ineligibility	1 (16.7)	Sample ineligibility	1 (16.7)
4	60	1	Air	1 (16.7)	Air	1 (16.7)
			Clot	0 (0)	Clot	0 (0)
			Sample ineligibility	1 (16.7)	Sample ineligibility	2 (33.3)
5	120	1	Air	1 (16.7)	Air	0 (0)
			Clot	0 (0)	Clot	0 (0)
			Sample ineligibility	1 (16.7)	Sample ineligibility	0 (0)
6	180	1	Air	2 (33.3)	Air	1 (16.7)
			Clot	0 (0)	Clot	0 (0)
			Sample ineligibility	0 (0)	Sample ineligibility	0 (0)
Occurrence rate out of 36 cases			Air	8 (21.6)	Air	5 (13.5)
			Clot	4 (10.8)	Clot	4 (10.8)
			Sample ineligibility	4 (10.8)	Sample ineligibility	3 (8.1)
			Error	1 (2.7)	Error	0 (0)

**Table 3 biomedicines-12-02421-t003:** Transfusion volume error rate and laboratory temperature for each experiment.

No	Anyfusion H-100		Terumo TE-LM700		No	Anyfusion H-100		Terumo TE-LM700	
	Error Rate (%)	Temperature (°C)	Error Rate (%)	Temperature (°C)		Error Rate (%)	Temperature (°C)	Error Rate (%)	Temperature (°C)
1	4.00	25.4	−4.00	24.3	30	−4.25	25.1	−1.41	25.1
2	6.33	25.6	−8.30	24.8	31	5.91	25	−3.16	24.8
3	−4.00	25	3.00	24.9	32	2.10	26.3	−3.30	24.9
4	5.67	24.9	−3.00	24.6	33	2.33	26.2	−2.67	25
5	2.67	25.3	−7.30	24.3	34	−1.22	24.3	−3.44	24
6	3.67	24.2	−4.33	24.1	35	Error		−4.33	24.6
7	−2.33	25.1	4.00	24.2	36	−3.77	24.9	3.78	24.6
8	1.67	25.3	−3.60	24.6	37	4.50	25.3	3.78	24.9
9	−7.00	24.6	−4.67	24.8	D1	−4.50	24.8	3.25	24.3
10	−5.67	24.8	−3.33	24.3	D2	−2.58	24.9	0.42	24.5
11	1.67	24.9	4.33	24.3	D3	−3.17	25	−4.58	24.3
12	3.33	25.1	2.00	24.3	D4	4.25	25.1	−3.41	24.5
13	−4.33	24.7	−6.00	24.5	D5	−5.08	24.7	−5.17	24.6
14	−0.06	24.9	−3.00	24.1	D6	0.67	24.6	6.33	24.1
15	4.33	25	−5.33	24.6	D7	4.58	24.8	7.00	24.2
16	5.67	24.4	−2.67	24.7	D8	2.25	24.9	2.08	25.1
17	3.33	25	−4.00	24.6	D9	−4.83	25.2	1.41	24.8
18	7.33	24.9	−1.33	24.4	D10	2.58	24.7	−1.75	24.6
19	5.00	24.5	−3.67	24.4	D11	3.67	24.3	−5.33	24.8
20	1.33	25.4	−4.67	24.4	D12	0.75	24.6	3.42	24.4
21	−4.83	24.7	−3.67	24.9	D13	−4.33	24.5	6.92	25
22	−3.17	25.3	−5.50	25.1	D14	−2.50	25	−3.17	24.6
23	−4.50	24.3	−5.10	24.2	D15	−4.00	24.7	−3.83	24.7
24	5.00	24.4	−4.80	24.4	D16	5.25	25.2	5.25	24.9
25	−4.67	24.4	−4.16	24.5	D17	6.50	24.4	−0.92	24.3
26	−2.75	26.2	5.08	24.2	D18	−3.83	24.7	5.41	24.6
27	−4.25	25.8	3.50	24.9	D19	−2.67	24.3	−6.00	24.2
28	−3.67	25.1	1.90	24.3	D20	4.17	25.3	5.25	24.8
29	3.91	25.4	−4.91	25.1	Average	3.77	25.0	4.00	24.6

D: dilution test.

**Table 4 biomedicines-12-02421-t004:** Difference in error occurrence through chi-square test between two medical devices.

Medical Device	Air Trapping (n, %)	Clot (n, %)	Normal (n, %)	X^2^	*p*-Value
Anyfusion H-100	9 (15.8)	4 (7.0)	44 (77.2)	0.294	0.863
Terumo TE-LM700	7 (12.3)	4 (7.0)	46 (80.7)		

**Table 5 biomedicines-12-02421-t005:** Comparison of the mean hemolysis rates using *t*-test between two medical devices.

Medical Device	Values (%, Mean ± SD)	*t*	*p*-Value
Anyfusion H-100	0.566 ± 0.095	0.294	0.863
Terumo TE-LM700	0.518 ± 0.126		

SD: standard deviation.

## Data Availability

The data that support the findings of this study are available on request from the corresponding author. The data are not publicly available due to privacy or ethical restrictions.

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
