# Peer review of "Comparative Analysis of Blood Transfusion Accuracy and Hemolysis Rate of Transfusion Cartridge Set Between Conventional Infusion Pumps and Cylinder-Type Infusion Pumps"

_biomedicines, 2024, doi:10.3390/biomedicines12112421_

Round 1

Reviewer 1 Report

Comments and Suggestions for Authors

This manuscript described a clinical study to compare the blood transfusion accuracy and hemolysis rate between the conventional pump, Terufusion TE-LM700, and the novel cylinder-type infusion pump, Anyfusion H-100. The method is sound.

A few comments:

1.     Is the Anyfusion H-100 pump already approved for clinical use? If so, the accuracy of the Anyfusion H-100 pump should have already been tested before approval. Why is this study necessary to do the test again? The necessity of this study should be more clearly described in the manuscript.

2.     Figure 1 just showed the pictures of two pumps. The machine’s appearance of two pumps did not provide valuable information for this study. It would be better to show the mechanism or functions of the two pumps in this figure.

3.     Figure 2 just showed the picture of multiple blood bags. It also did not provide any valuable information. It would be better to show the study design, for example, the designed scenarios and workflow of sampling in this figure.  

4.     This study is a clinical study. Why the title is “simulation analysis”?

Author Response

Comments and Suggestions for Authors

This manuscript described a clinical study to compare the blood transfusion accuracy and hemolysis rate between the conventional pump, Terufusion TE-LM700, and the novel cylinder-type infusion pump, Anyfusion H-100. The method is sound.

A few comments:

  1. Is the Anyfusion H-100 pump already approved for clinical use? If so, the accuracy of the Anyfusion H-100 pump should have already been tested before approval. Why is this study necessary to do the test again? The necessity of this study should be more clearly described in the manuscript.

☞ Thank you for your good opinion. We fully understood that the medical device should test before approval. First of all, we apologize for the confusion caused to the reviewer. To answer the conclusion, we conducted a study that tested the device before approval.

The blood transfusion cartridge set mounted on the cylinder-type infusion pump that was developed in this study was approved on January 13, 2021 (Attachment 1 Below). We also mentioned this study's IRB approval (CR320345) in the manuscript, and the approval date was June 30, 2020 (Attachment 2 Below). After that, we received a total of 57 research-grade blood products from the Korean Red Cross Blood Services from October 8, 2020, to December 10, 2020 (Attachment 3). (The documents were translated from the Korean to English using Google Translate.)

☞ Finally, we can briefly summarize it in a timetable.

  • June 30, 2020: IRB approval (No.: CR320345)
  • August 10, 2020: Request for the research-grade blood products (Wonju Severance Christian Hospital -> Korea Red Cross)
  • October 8, 2020 ~ December 10, 2020: Provision of the research-grade blood products (Korea Red Cross -> Wonju Severance Christian Hospital)
  • January 13, 2021: Approval of medical device product from the Ministry of Food and Drug Safety
  1. Figure 1 just showed the pictures of two pumps. The machine’s appearance of two pumps did not provide valuable information for this study. It would be better to show the mechanism or functions of the two pumps in this figure.

☞ Thank you for pointing out the lack of valuable information about the device. We tried to improve the paragraph by adding a description of the development of the infusion pump in the Introduction section and the working mechanism of the cylinder cartridge in the Materials and Methods section.

☞ [Before]

(Line 68-81) A method to improve this situation is to integrate the syringe injection method with the infusion method, and the Anyfusion H-100 device has been developed to achieve this. Anyfusion H-100 operates on the principle of two pistons attached to two disks within a donut-shaped cylinder, which rotate via two motors to aspirate and dispense liquids (medications), gases, and flowable solids (food, blood) regardless of gravity or position [6].

In this study, we aimed to verify whether the drug delivery system, Anyfusion H-100, can accurately administer medications used for infants and neonates, and whether it can quickly and precisely deliver large amounts of fluids in emergency situations. Using research-grade blood, we tested if the blood could be infused at an accurate volume and rate, and examined hemolysis levels before and after blood transfusion to confirm its viability for transfusion. Through this, we aimed to demonstrate that this new type of infusion pump has no functional differences compared to conventional infusion pumps and to prove its suitability as a medical device.

[Revised]

(Line 68-81) To improve the limitations of the structure of conventional peristaltic and syringe-type infusion pumps, it was developed a cylinder-type infusion pump (Anyfusion H-100) that integrates the syringe infusion method with the infusion method (Figure 1). This device operates on the principle of two pistons attached to two disks within a donut-shaped cylinder, which rotates through two motors to aspirate and dispense liquids (drugs), gases, and fluid solids (food, blood) regardless of gravity or position [6].

In this study, using research-grade blood products, we tested if the blood could be infused at an accurate volume and rate, and examined hemolysis levels before and after blood transfusion to confirm its viability for transfusion. Through this, we aimed to demonstrate that this new type of infusion pump has no functional differences compared to conventional infusion pumps and to prove its suitability as a medical device.

(Line 89-106) 2.1. Mechanism of cylinder-type infusion pump

The cylinder-type infusion pump has two unique characteristics in terms of flow accuracy and stability by using a dedicated cylinder cartridge including a new operating principle using high-precision motor control and an automatic locking system to completely fix the cartridge to the pump body. The core mechanism of the device is the continuous cross-cycling of two pistons inside the dedicated donut-shaped cylinder cartridge, with independent motor control for each piston. This mechanism converts the linear motion of one piston inside a conventional syringe into the rotary motion of two pistons inside a spherical cylinder. In the donut-shaped cylinder, each piston is programmed to rotate in a different direction from the others, allowing for precise control of the fluid infusion. The piston between the fluid inlet and outlet has a fixed shaft (S or Sʼ), while the other piston has a rotating shaft (R or Rʼ). Under precise control by an independent motor according to a set injection rate, the rotating pistons (R or Rʼ) rotate counterclockwise, so that a precise amount of fluid is simultaneously aspirated from the inlet and expelled from the outlet (Figure 2).

  1. Figure 2 just showed the picture of multiple blood bags. It also did not provide any valuable information. It would be better to show the study design, for example, the designed scenarios and workflow of sampling in this figure.

☞ Thank you for your good comments. According to your opinion, we added a flow chart and sentences to the '2.3. Study protocols' paragraph after the figure of the research-grade blood products.

☞ [Before] (Line 112-114)

2.2. Study protocols

  1. Quantification and Error Rate According to Infusion Rate and Time
  2. Degree of Hemolysis According to Infusion Rate

 [Revised] (Line 131-143)

2.3. Study protocols

The values ​​measured between the two devices were compared using the research blood products provided as follows:

  1. Quantification and Error Rate According to Infusion Rate and Time
  2. Degree of Hemolysis According to Infusion Rate

In the non-dilution trials, the blood sample infusion volume per hour was set to 10 cc/hr (180 min), 15 cc/hr (120 min), 30 cc/hr (60 min), 60 cc/hr (60 min), 120 cc/hr (60 min), and 180 cc/hr (60 min). In addition, in the dilution trials, the blood sample infusion volume per hour was set to 60 cc/hr (120 min), 120 cc/hr (60 min), 180 cc/hr (40 min), and 240 cc/hr (30 min) (Figure 5).

  1. This study is a clinical study. Why the title is “simulation analysis”?

☞ Thank you for pointing out the inappropriate title. We changed the title to represent it more clearly.

☞ [Before] (Line 2-4) Simulation analysis of blood transfusion accuracy and hemolysis rate of conventional infusion pump and cylinder-type infusion pump

   [Revised] (Line 2-4) Comparative Analysis of Blood Transfusion Accuracy and Hemolysis Rate of Transfusion Cartridge Set between Conventional Infusion Pump and Cylinder-type Infusion Pump 

Reviewer 2 Report

Comments and Suggestions for Authors

Comments to Authors:

Abstract Section:

1) At the background effectively sets the stage for the study. It might be beneficial to briefly mention why hemolysis is particularly concerning in neonates, as this will underscore the relevance of your research study!

2) All keywords should be checked and formatted according to MeSH (Medical Subject Headings) standards.

Introduction Section:

1) In my opinion, it was better use a brief summary of the challenges faced with traditional pumps before presenting your innovative solution would help contextualize your study's significance.

2) The some reference was not supported your text. I am sure, you missed some of references on your text body!

Please check:

Lines 34-39. 

Lines 50-64. 

3) Lines 58-62: You mention issues such as backflow, startup delays, and flow variations. Providing specific examples or data on how these issues impact patient outcomes would strengthen your argument for the need for improved devices.

4) Some technical terms (e.g., "peristaltic motion," "syringe-based infusion pumps") may not be familiar to all readers. Consider defining these terms briefly to ensure clarity.

Materials and Methods Section:

1) The 1:1 dilution method with normal saline is mentioned, but further clarification on the rationale behind this choice and its relevance to emergency blood transfusions would enhance understanding!?

2) Consider discussing any potential confounding factors that could influence the results, such as patient Materials and Methods or pre-existing conditions!?

Results Section:

Among the things that are ambiguous in the analysis of this manuscript:

1) Table 4, since it compares the error ratio of the two methods, it was reasonable to make this comparison for each step, which includes the steps: Air trapping, Clot, and Normal.

2- In Table 5, the average comparison of the two methods has been done, which is not logical with the chi-square test. Rather, it should have been done with the t-test method if it was normal.

Discussion Section:

1) In my opnion, suggestions for future research could be included in the text. What specific aspects of infusion technology or methodologies require further investigation based on the findings of this study?

Conclusion Section: 

Recognize any limitations of the study in the conclusion would provide a more balanced view. What aspects might affect the generalizability of the results?

Comments on the Quality of English Language

 The English could be improved to more clearly express the research.

Author Response

Comments and Suggestions for Authors

Abstract Section:

1) At the background effectively sets the stage for the study. It might be beneficial to briefly mention why hemolysis is particularly concerning in neonates, as this will underscore the relevance of your research study!

☞ Thank you for your comment with your suggestions. This manuscript made a discussion the development of a cylindrical infusion pump to improve the problems of conventional infusion pumps such as hemolysis, flow variation, and infection risk, and analyzed the accuracy and hemolysis rate according to infusion rate and time using a transfusion cartridge set. We felt that the mention of neonates was inappropriate, so we tried to modify the paragraph flow to make it smooth.

☞ [Before] (Line 14-19) Infusion pumps are critical in clinical settings, precisely delivering fluids such as medications and blood products. However, maintaining accuracy and preventing hemolysis is essential when administering blood, particularly to neonates. Conventional pumps may cause hemolysis due to pressure variations during transfusion. This study compares the blood transfusion accuracy and hemolysis rate between the conventional pump, Terufusion TE-LM700 (Terufusion) and the novel cylinder-type infusion pump, Anyfusion H-100 (Anyfusion);

[Revised] (Line 15-22) Background/Objectives: Infusion pumps are critical in delivering fluids, including medications and blood products, in controlled amounts. However, conventional pumps can cause hemolysis and other issues such as flow variations and infection risks, especially during blood transfusions. To address these limitations, a novel cylinder-type infusion pump, the Anyfusion H-100, was developed, which includes a specialized blood transfusion cartridge set that combines syringe and peristaltic infusion methods. This study evaluates the accuracy and hemolysis rates of the Anyfusion H-100 compared to conventional pumps, aiming to confirm its viability as a safe and effective medical device for blood transfusions;

2) All keywords should be checked and formatted according to MeSH (Medical Subject Headings) standards.

☞ We have modified the keywords according to your advice to conform to the MeSH (Medical Subject Headings) standard (https://www.ncbi.nlm.nih.gov).

☞ [Before] (Line 31) Keywords: Infusion pump; Cylinder-type; Hemolysis; Blood transfusion

[Revised] (Line 33) Keywords: Infusion Pump; Cylinder-type; Hemolysis; Blood Transfusion 

Introduction Section:

1) In my opinion, it was better use a brief summary of the challenges faced with traditional pumps before presenting your innovative solution would help contextualize your study's significance.

☞ Thank you for your sincere advice. In the Introduction section, we mentioned the problems of conventional infusion pumps (hemolysis, infection risk, flow variation, tubing recoil). Also, we tried to contextualize this by developing a cylinder-type infusion pump.

☞ [Before] (Line 50-68) Generally, infusion pumps for medications inject a predetermined volume into the patient based on a program that rotates a motor. When using an infusion pump for blood transfusions, a constant pressure is applied, which may cause physical damage to red blood cells as they pass through the needle, increasing the risk of hemolysis. Therefore, when transfusing red blood cells or other blood products using an infusion pump, it is crucial to verify that the device is suitable for its intended purpose, especially since visible hemolysis can occur with red blood cell products. In the case of syringe-based infusion pumps, frequent syringe replacements pose a risk of infection, necessitating careful attention and leading to increased workload. Additionally, issues associated with the operation of traditional infusion pumps, such as infusion accuracy, backflow, and startup delays, can become problematic during low-volume transfusions. Conventional infusion pumps control flow through the peristaltic motion of tubing, which can result in flow variations of ±5% to 20% per hour, depending on the tubing material and elasticity. When issues like tubing recoil arise, the system may need to be reset or the tubing replaced. Furthermore, the drive mechanism that compresses the tubing may also lead to hemolysis.

Syringe pumps used for precise infusion have compatibility issues due to varying syringe specifications among manufacturers, and they require frequent resetting when large volumes are injected. The process of transferring blood into a syringe for precise infusion increases vulnerability to contamination and infection [2,5].

[Revised] (Line 52-70) Generally, infusion pumps for medications inject a predetermined volume into the patient based on a program that rotates a motor. When using an infusion pump for blood transfusions, a constant pressure is applied, which may cause physical damage to red blood cells as they pass through the needle, increasing the risk of hemolysis [7]. Therefore, when transfusing red blood cells or other blood products using an infusion pump, it is crucial to verify that the device is suitable for its intended purpose, especially since visible hemolysis can occur with red blood cell products [8]. In the case of syringe-based infusion pumps, frequent syringe replacements pose a risk of infection, necessitating careful attention and leading to increased workload. Additionally, issues associated with the operation of traditional infusion pumps, such as infusion accuracy, backflow, and startup delays, can become problematic during low-volume transfusions. Even with this issue, small dosing errors in these vasoactive drugs can result in severe hemodynamic fluctuations in vulnerable patients [9-10]. Conventional infusion pumps control flow through the peristaltic motion of tubing, which can result in flow variations of ±5% to 20% per hour, depending on the tubing material and elasticity [11]. When issues like tubing recoil arise, the system may need to be reset or the tubing replaced. Furthermore, the drive mechanism that compresses the tubing may also lead to hemolysis. Syringe pumps used for precise infusion have compatibility issues due to varying syringe specifications among manufacturers, and they require frequent resetting when large volumes are injected. The process of transferring blood into a syringe for precise infusion increases vulnerability to contamination and infection [4,12].

2) The some reference was not supported your text. I am sure, you missed some of references on your text body!

Please check:

Lines 34-39. 

Lines 50-64.

☞ Thank you for pointing out the lack of references. We tried to add the relevant references to support the text body.

☞ [Before]

(Line 34-39) Infusion pumps are medical devices used to deliver fluids, such as nutrients and medications, into a patient's body in controlled amounts. The evolution of infusion pumps from basic mechanical devices in the 1970s and 1980s to the advanced, multifunctional devices of today is well-documented. Modern infusion pumps incorporate various features and alarms designed to alert users about issues such as nearing completion of infusions or the need for attention due to detected anomalies [1].

(Line 50-64) Generally, infusion pumps for medications inject a predetermined volume into the patient based on a program that rotates a motor. When using an infusion pump for blood transfusions, a constant pressure is applied, which may cause physical damage to red blood cells as they pass through the needle, increasing the risk of hemolysis. Therefore, when transfusing red blood cells or other blood products using an infusion pump, it is crucial to verify that the device is suitable for its intended purpose, especially since visible hemolysis can occur with red blood cell products. In the case of syringe-based infusion pumps, frequent syringe replacements pose a risk of infection, necessitating careful attention and leading to increased workload. Additionally, issues associated with the operation of traditional infusion pumps, such as infusion accuracy, backflow, and startup delays, can become problematic during low-volume transfusions. Conventional infusion pumps control flow through the peristaltic motion of tubing, which can result in flow variations of ±5% to 20% per hour, depending on the tubing material and elasticity. When issues like tubing recoil arise, the system may need to be reset or the tubing replaced. Furthermore, the drive mechanism that compresses the tubing may also lead to hemolysis.

[Revised]

(Line 36-41) Infusion pumps are medical devices used to deliver fluids, such as nutrients and medications, into a patient's body in controlled amounts [1]. The evolution of infusion pumps from basic mechanical devices in the 1970s and 1980s to the advanced, multifunctional devices of today is well-documented [2]. Modern infusion pumps incorporate various features and alarms designed to alert users about issues such as nearing completion of infusions or the need for attention due to detected anomalies [3].

(Line 52-68) Generally, infusion pumps for medications inject a predetermined volume into the patient based on a program that rotates a motor. When using an infusion pump for blood transfusions, a constant pressure is applied, which may cause physical damage to red blood cells as they pass through the needle, increasing the risk of hemolysis [7]. Therefore, when transfusing red blood cells or other blood products using an infusion pump, it is crucial to verify that the device is suitable for its intended purpose, especially since visible hemolysis can occur with red blood cell products [8]. In the case of syringe-based infusion pumps, frequent syringe replacements pose a risk of infection, necessitating careful attention and leading to increased workload. Additionally, issues associated with the operation of traditional infusion pumps, such as infusion accuracy, backflow, and startup delays, can become problematic during low-volume transfusions. Conventional infusion pumps control flow through the peristaltic motion of tubing, which can result in flow variations of ±5% to 20% per hour, depending on the tubing material and elasticity [9]. When issues like tubing recoil arise, the system may need to be reset or the tubing replaced. Furthermore, the drive mechanism that compresses the tubing may also lead to hemolysis.

3) Lines 58-62: You mention issues such as backflow, startup delays, and flow variations. Providing specific examples or data on how these issues impact patient outcomes would strengthen your argument for the need for improved devices.

☞ Thank you for your opinion. We added a sentence related to the issue's impact on patient outcomes.

☞ [Before] (Line 58-62) Additionally, issues associated with the operation of traditional infusion pumps, such as infusion accuracy, backflow, and startup delays, can become problematic during low-volume transfusions. Conventional infusion pumps control flow through the peristaltic motion of tubing, which can result in flow variations of ±5% to 20% per hour, depending on the tubing material and elasticity.

[Revised] (Line 60-66) Additionally, issues associated with the operation of traditional infusion pumps, such as infusion accuracy, backflow, and startup delays, can become problematic during low-volume transfusions. Even with this issue, small dosing errors in these vasoactive drugs can result in severe hemodynamic fluctuations in vulnerable patients [9-10]. Conventional infusion pumps control flow through the peristaltic motion of tubing, which can result in flow variations of ±5% to 20% per hour, depending on the tubing material and elasticity [11].

4) Some technical terms (e.g., "peristaltic motion," "syringe-based infusion pumps") may not be familiar to all readers. Consider defining these terms briefly to ensure clarity.

 ☞ Thank you for your good comments. We tried to be familiar to all readers as adding the paragraph about the mechanism of cylinder-type infusion pump..

☞ [Revised] (Line 96-113) 2.1. Mechanism of cylinder-type infusion pump

The cylinder-type infusion pump has two unique characteristics in terms of flow accuracy and stability by using a dedicated cylinder cartridge including a new operating principle using high-precision motor control and an automatic locking system to completely fix the cartridge to the pump body. The core mechanism of the device is the continuous cross-cycling of two pistons inside the dedicated donut-shaped cylinder cartridge, with independent motor control for each piston. This mechanism converts the linear motion of one piston inside a conventional syringe into the rotary motion of two pistons inside a spherical cylinder. In the donut-shaped cylinder, each piston is programmed to rotate in a different direction from the others, allowing for precise control of the fluid infusion. The piston between the fluid inlet and outlet has a fixed shaft (S or Sʼ), while the other piston has a rotating shaft (R or Rʼ). Under precise control by an independent motor according to a set injection rate, the rotating pistons (R or Rʼ) rotate counterclockwise, so that a precise amount of fluid is simultaneously aspirated from the inlet and expelled from the outlet (Figure 2).

Materials and Methods Section:

1) The 1:1 dilution method with normal saline is mentioned, but further clarification on the rationale behind this choice and its relevance to emergency blood transfusions would enhance understanding!?

☞ Thank you for your good opinion. We added a sentence and a reference that noted the relationship between saline dilution and emergency transfusion.

☞ [Before] (Line 83-88) The experiment was conducted with six different infusion rates: 10cc per hour for 3 hours, 15cc per hour for 2 hours, 30cc per hour for 1 hour, 60cc per hour for 1 hour, 120cc per hour for 1 hour, and 180cc per hour for 1 hour. Each experiment was repeated six times. Additionally, 20 tests were performed using a 1:1 dilution method with normal saline to simulate emergency blood transfusions. All experiments were carried out with an experimental group (Anyfusion H-100) and a control group (Terufusion TE-LM700).

   [Revised] (Line 87-95) The experiment was conducted with six different infusion rates: 10cc per hour for 3 hours, 15cc per hour for 2 hours, 30cc per hour for 1 hour, 60cc per hour for 1 hour, 120cc per hour for 1 hour, and 180cc per hour for 1 hour. Each experiment was repeated six times. Additionally, 20 tests were performed using a 1:1 dilution method with normal saline to simulate emergency blood transfusions. Normal saline was chosen because it prevents hemolysis, provides volume expansion, and is fully compatible with blood products, making it a suitable choice for rapid transfusion under emergency conditions [14]. All experiments were carried out with an experimental group (Anyfusion H-100) and a control group (Terufusion TE-LM700).

2) Consider discussing any potential confounding factors that could influence the results, such as patient Materials and Methods or pre-existing conditions!?

☞ Thank you for your valuable feedback. We fully understand your opinions and since this study is a comparative evaluation of devices rather than patients, it may be worth considering in further studies. We think there are many variables such as patient age, sex, disease, and health status, so if the control and experimental groups in a clinical trial are similar in age, sex, and health status, it would be worthwhile as a further clinical study. We appreciate your concerns and will consider them in further research including clinical trials. 

Results Section:

Among the things that are ambiguous in the analysis of this manuscript:

1) Table 4, since it compares the error ratio of the two methods, it was reasonable to make this comparison for each step, which includes the steps: Air trapping, Clot, and Normal.

☞ We intended that Table 4 was analyzed to prove that there is no difference between normal and abnormal transfusion between existing products that are generally used and certified as medical devices and new cylinder-based infusion pumps. Air trapping and clots are expressed separately only to express the problems in detail. From the authors' perspective, we thought that presenting it as it was in Table 4 would convey the meaning better to the reader. Nevertheless, if you request a revision to this table, we will try to rewrite Table 4 by dividing it into normal and abnormal categories.

2) In Table 5, the average comparison of the two methods has been done, which is not logical with the chi-square test. Rather, it should have been done with the t-test method if it was normal.

☞ Thank you for pointing out our mistake. The X2 in Table 5 is a typo. We actually performed a t-test and described the results. We changed the word “X2” to “t”.

Discussion Section:

1) In my opinion, suggestions for future research could be included in the text. What specific aspects of infusion technology or methodologies require further investigation based on the findings of this study?

☞ Thank you for your opinion. We have tried to suggest topics for future research after considering your question.

☞ [Before] (Line 228-230) Based on the results of this study and a review of previous research, it is determined that the Anyfusion H-100 can sufficiently fulfill its role as an infusion pump capable of performing blood transfusions

[Revised] (Line 263-268) Based on the results of this study and a review of previous research, it is determined that the Anyfusion H-100 can sufficiently fulfill its role as an infusion pump capable of performing blood transfusions. In further researches, it may be suggested to conduct additional clinical trials on the topic of clinical benefits that utilize the advantages of this cylinder pump to reduce hemodynamic instability in patients or reduce the clinical workload of medical staff.

Conclusion Section: 

Recognize any limitations of the study in the conclusion would provide a more balanced view. What aspects might affect the generalizability of the results?

☞ Thank you for your good comments. We mentioned the limitations of this study in the Conclusion section .

☞ [Before] (Line 242-245) Therefore, the Anyfusion H-100, as an infusion pump capable of performing blood transfusions, is expected to be highly useful as a medical device that integrates the syringe infusion method and the infusion method, effectively addressing each method's functional issues.

   [Revised] (Line 280-285) However, this study has limitations in that it was a laboratory study using research-grade blood products and safety was not assessed as no actual patients were transfused using these devices. Nevertheless, the Anyfusion H-100, as an infusion pump capable of performing blood transfusions, is expected to be highly useful as a medical device that integrates the syringe infusion method and the infusion method, effectively addressing each method's functional issues.

Round 2

Reviewer 1 Report

Comments and Suggestions for Authors

Thank you for your response. I am satisfied with this revised version.